# Functional Delineation of a Protein–Membrane Interaction Hotspot Site on the HIV-1 Neutralizing Antibody 10E8

**DOI:** 10.3390/ijms231810767

**Published:** 2022-09-15

**Authors:** Sara Insausti, Miguel Garcia-Porras, Johana Torralba, Izaskun Morillo, Ander Ramos-Caballero, Igor de la Arada, Beatriz Apellaniz, Jose M. M. Caaveiro, Pablo Carravilla, Christian Eggeling, Edurne Rujas, Jose L. Nieva

**Affiliations:** 1Instituto Biofisika (CSIC-UPV/EHU), University of the Basque Country (UPV/EHU), 48080 Bilbao, Spain; 2Department of Biochemistry and Molecular Biology, University of the Basque Country (UPV/EHU), 48080 Bilbao, Spain; 3Department of Physiology, Faculty of Pharmacy, University of the Basque Country (UPV/EHU), Paseo de la Universidad, 7, 01006 Vitoria-Gasteiz, Spain; 4Laboratory of Global Healthcare, School of Pharmaceutical Sciences, Kyushu University, Fukuoka 819-0395, Japan; 5Leibniz Institute of Photonic Technology e.V., 07745 Jena, Germany; 6Faculty of Physics and Astronomy, Institute of Applied Optics and Biophysics, Friedrich Schiller University Jena, 07743 Jena, Germany; 7Medical Research Council Human Immunology Unit, Weatherall Institute of Molecular Medicine, University of Oxford, Oxford OX1 2JD, UK; 8Ikerbasque, Basque Foundation for Science, 48013 Bilbao, Spain; 9Pharmacokinetic, Nanotechnology and Gene Therapy Group, Faculty of Pharmacy, University of the Basque Country UPV/EHU, 01006 Vitoria-Gasteiz, Spain; 10Microbiology, Infectious Disease, Antimicrobial Agents, and Gene Therapy, Bioaraba, 01006 Vitoria-Gasteiz, Spain

**Keywords:** MPER epitope recognition, protein–membrane interactions, antibody–membrane interactions, anti-MPER HIV antibody, HIV neutralization, broadly neutralizing antibody 10E8

## Abstract

Antibody engagement with the membrane-proximal external region (MPER) of the envelope glycoprotein (Env) of HIV-1 constitutes a distinctive molecular recognition phenomenon, the full appreciation of which is crucial for understanding the mechanisms that underlie the broad neutralization of the virus. Recognition of the HIV-1 Env antigen seems to depend on two specific features developed by antibodies with MPER specificity: (i) a large cavity at the antigen-binding site that holds the epitope amphipathic helix; and (ii) a membrane-accommodating Fab surface that engages with viral phospholipids. Thus, besides the main Fab–peptide interaction, molecular recognition of MPER depends on semi-specific (electrostatic and hydrophobic) interactions with membranes and, reportedly, on specific binding to the phospholipid head groups. Here, based on available cryo-EM structures of Fab–Env complexes of the anti-MPER antibody 10E8, we sought to delineate the functional antibody–membrane interface using as the defining criterion the neutralization potency and binding affinity improvements induced by Arg substitutions. This rational, Arg-based mutagenesis strategy revealed the position-dependent contribution of electrostatic interactions upon inclusion of Arg-s at the CDR1, CDR2 or FR3 of the Fab light chain. Moreover, the contribution of the most effective Arg-s increased the potency enhancement induced by inclusion of a hydrophobic-at-interface Phe at position 100c of the heavy chain CDR3. In combination, the potency and affinity improvements by Arg residues delineated a protein–membrane interaction site, whose surface and position support a possible mechanism of action for 10E8-induced neutralization. Functional delineation of membrane-interacting patches could open new lines of research to optimize antibodies of therapeutic interest that target integral membrane epitopes.

## 1. Introduction

The broadly neutralizing HIV-1 antibodies (bnAbs) isolated to date constitute useful tools, not only in the development of potential candidate vaccines and therapeutics, but also in structure–function studies aimed at elucidating the mechanism of viral entry mediated by the envelope glycoprotein (Env) [1,2,3,4,5,6]. Among the broadest identified so far, those targeting a helical Env epitope at the C-terminus of the conserved membrane-proximal external region (MPER) compose a distinct class of HIV-1 bnAbs [7,8,9,10,11,12,13]. The antibody 10E8, the best-characterized member of this class, has garnered attention for its extraordinary neutralization breadth; additionally, it has been a major focus for research regarding antibody optimization and interactions near the viral membrane [8,14,15,16,17,18]. Consequently, 10E8 was incorporated in multi-specific antibody formats as a strategy to develop therapeutic candidates capable of achieving pan-neutralization of multiple HIV-1 isolates [19,20,21,22,23,24,25]. Supporting its therapeutic potential, the passive administration of 10E8 provides protection against infection in animal models [26]. In addition, the expression of glycosylphosphatidylinositol (GPI)-anchored 10E8 also appears to render cells resistant to HIV-1 infection, providing new strategies based on this antibody to treat chronic infection [27].

X-ray diffraction structural studies carried out concomitant to its identification and isolation in 2012 [8] revealed that 10E8 engages with an α-helix beginning at ca. N671 and ending at R683, the last MPER residue belonging to the Env ectodomain. Further analyses revealed that the 10E8 epitope and its connection to the gp41 transmembrane domain (TMD) can form a continuous, straight helix [16]. The orientation of the Fab based on the position of phospholipid and phosphate-binding sites at the antibody surface supported an interaction model according to which the continuous MPER-TMD helix emerges obliquely from the HIV membrane, whereas the Fab approaches laterally to the epitope protruding from the membrane plane [16,17]. This binding model is also consistent with the data derived from cryogenic electron microscopy (cryo-EM) of the Env–Fab complexes reconstituted in detergent micelles and lipid nanodiscs [28,29].

The available evidence also suggests that the effective molecular recognition of MPER by the Fab 10E8 requires its association with the viral membrane interface through an ‘accommodation surface’ [5,16,17,18,28,29,30,31,32,33]. Several observations highlight the potential relevance of these interactions for the antibody function. An increase in the net negative charge by introduction of Asp or Glu at membrane-facing/solvent-exposed positions within the CDRs 1 and 2 and FRs 1 and 3 of the Fab light chain (LC) was shown to decrease the neutralization potency of the antibody [17]. Consistent with this observation, we demonstrated that the addition of a positive charge by introducing three additional Arg residues at the LC regions, CDR1, CDR2 and FR3 (3R mutant), enhanced both the 10E8 potency (by ca. 5–10-fold) and the affinity for the native Env in the virions [5,30]. N-glycan additions and Gly substitutions also pinpointed the residues located at the CDR1, CDR2, FR1 and FR3 of the LC as facing the viral membrane and mediating the neutralization function of the antibody [18]. On the other hand, a single substitution of the membrane-facing/solvent-exposed S100c residue at the heavy chain (HC) CDR3 by aromatics rendered a 10E8 antibody (100cF mutant) with a neutralization potency enhanced by ca. 8–10-fold [18]. In addition, consistent with this observation, the inclusion of aromatic residues at more remote LC positions of the membrane-accommodating surface displayed a favorable effect on the 10E8-mediated neutralization and enhanced binding to Env [31,32,33]. 

Despite the great deal of available structural and functional data, the molecular mechanism underlying the effective MPER recognition by 10E8 in virions is not firmly established and different views persist to date [6,29,34,35,36]. One approach which may shed light on the process, is to determine the requirements that mutations must fulfill to improve the biological function (neutralization and affinity) of the antibody. Thus, we analyze the membrane-facing surfaces inferred from available structural information to establish: (i) the requirements (effective positions and combinations) for the functional improvement of 10E8 by the Arg residues that promote the Fab–membrane interactions, and (ii) their ability to supplement the potentiation induced by Phe at position 100c of the HC. Collectively, our neutralization and affinity data support the existence of a protein–membrane interaction “hotspot”, whose surface and position correlate with a defined 10E8 orientation with respect to the Env ectodomain, MPER helix and viral membrane surface. This finding might be useful in better defining the mechanisms of action for this class of broadly neutralizing HIV antibodies.

## 2. Results

### 2.1. Inclusion of Arg Residues in the Surface of the Constant Domain That Can Accommodate the Membrane Does Not Improve Antibody Potency

Consistent with the high degree of flexibility reported for lambda LC Fabs [37], a comparison of available 10E8 crystal structures reflects a full range of elbow angles, ranging from small (119°) to very large values (223°) (Figure 1A, left). Aiming to identify all of the possible membrane-facing areas existing in the Fab surface, we compared the three-dimensional structures after orienting the antibody based on the position of the bound phosphates [16] and phospholipids [17] revealed by some solved crystals (Figure 1A, right). Based on this analysis, in the 5GHW structure with a low elbow angle and containing bound phosphates, both the constant and variant domains of the Fab occupy a position close to the membrane supporting the potential role of these regions in promoting epitope recognition within the membrane environment. 

The fitting of 10E8 Fabs solved with two different elbow angles into the cryo-EM map of an antibody–Env complex [28] further supported the membrane contact of this structure through both domains (Figure 1B). Again, the Fab structure 5GHW displayed both the constant (C_L_, C_H_) and variable (V_L_, V_H_) domains close to the position of the membrane surface. In contrast, the Fab structure 4G6F with a larger elbow angle, displayed the constant domain far from the membrane surface, and appeared to rotate through its pseudo two-fold main axis. Previously described Arg substitutions that aimed to potentiate antibody functional activity partially span both surfaces [30]. Hence, we used antibody potency enhancement as a read-out to map the functionally effective interacting regions within these potential membrane accommodation areas of 10E8.

In contrast to the previously observed 10E8 neutralization potentiation upon inclusion of Arg-s in the variable domain [30], promoting the electrostatic interactions through the constant domain did not result in a neutralization enhancement of the antibody (Figure 1C). No significant increase in the neutralization potency was observed when these mutations were included alone or in combination with the previously optimized 10E8-3R Fab (Figure 1C). Thus, even though Fab flexibility enables the accommodation of the membrane through a surface in the constant domain, this capability does not appear to contribute to the antiviral activity of the antibody. 

### 2.2. Requirements for Arg-Mediated 10E8 Potentiation and Complementation with HC-Ser100_c_Phe Mutation

Next, we studied the traits (positions and combinations) of the Arg-s introduced at the variable domain of the 3R mutant, which are required to potentiate the antibody. We generated single and double Arg mutants, and their potency was compared in pseudovirus (PsV) neutralization assays (Figure 2 and Appendix A (Appendix A)). Adding single Arg residues at LC positions S30R, N52R or S67R was not sufficient to improve the ability of the Fab to block viral entry (Appendix A). Similarly, the double combinations S30R/S67R or N52R/S67R, involving the LC FR3 residue S67, did not result in any appreciable effect in the neutralization potency of 10E8 (Figure 2A,B and Appendix A). In contrast, the double combination of the residues placed at CDRs 1 and 2, S30 and N52, respectively (S30R/N52R; henceforth designated as 2R) enhanced the 10E8 potency to the same extent as the triple combination S30R/N52R/S67R (3R) (Figure 2A,B and Appendix A). The activity and structural integrity of these combined, mutant Fabs were confirmed by enzyme-linked immunosorbent assay (ELISA) and circular dichroism (CD), respectively (Appendix A).

The similar contribution of the double and triple Arg substitutions to the antibody functional improvement was also observed in experiments using the 100cF 10E8 mutant, optimized by addition of hydrophobicity-at-interface at position 100c of the HCDR3 [18,39]. The combination of 100cF with either 2R or 3R resulted in properly folded antibodies displaying a potency increased by approximately 50-fold in comparison with the parental Fab 10E8-WT (Figure 2C,D, Appendix A). Thus, the mutations 2R and 100cF mapping to the membrane-facing surface of the Fab, appear to significantly improve neutralization at positions outside the specificity binding pocket by a combination of electrostatic interactions with membranes and interfacial hydrophobicity. In addition, the contribution of the S67R substitution to potency improvement appears to be less evident in this set up.

### 2.3. LC Arg and HC S100cF Substitutions Promote Antibody–Membrane Interaction

To obtain insights into the mechanisms of 2R and 2R-100cF (hereafter referred as 2RF) potentiation, we next measured the binding affinity of the Fab 10E8 mutants. First, we assessed the capacity of each Fab (WT, 2R and 2RF) to directly bind to the lipid membranes. To that end, the binding of the fluorescently labeled Fabs was quantitatively analyzed by confocal fluorescence microscopy of single lipid vesicles (giant unilamellar vesicles, GUVs) [32]. The fluorescence intensity changes determined at the lipid bilayer section (Figure 3A), indicated that the Fabs did not partition from water into vesicles made of the zwitterionic phospholipid 1-palmitoyl-2-oleoylphosphatidylcholine (POPC). Upon inclusion of 10 mole % of anionic 1-palmitoyl-2-oleoylphosphatidylserine (POPS) in the lipid composition, Fabs 2R and 2RF were detected at the membrane of the vesicles (Figure 3B and Appendix A). In accordance with a partitioning process mediated by electrostatic interactions, the amount of vesicle-bound Fab increased significantly in the GUVs that contained 50 mole % of POPS. Moreover, the amount of membrane-bound Fab detected in 2RF was higher than in the 2R samples, suggesting that the combination of charge and hydrophobicity elicited the process.

We next tested whether this spontaneous and independent association of Fabs with lipid membranes, i.e., in the absence of MPER epitope recognition, could contribute to viral neutralization in a process driven by increased avidity (Figure 3C). Thus, we generated a 10E8 IgG with the 2RF mutations and scored the reduction in the neutralization IC_50_ nM value in comparison with that obtained using the single Fab 10E8 (Figure 3D). The neutralization assays revealed similar degrees of potentiation for both of the antibody formats upon inclusion of the mutations (by a factor of ca. 50 in both cases). This apparent lack of avidity observed for the IgG 2RF indicates that, as previously suggested [5], the mutations improve neutralization through protein–membrane interactions that evolve linked to, but not in advance of, the antigen recognition process.

### 2.4. 10E8 Substitutions Promote MPER-Binding in Model Membranes and Native Environments

To obtain insights into the effect of the mutations after engaging with the MPER epitope, we next tested the binding of the Fabs to the MPER epitope peptides and native Env on virions. In a first approximation, a peptide that combined MPER with the N-terminus of the TMD was used as a Fab ligand to determine the effects on binding energetics by isothermal titration calorimetry (ITC). Peptide solubilization was achieved using the detergent DPC. The secondary structure determination monitored by CD revealed that the peptide adopted a helical conformation at a dodecylphosphocholine (DPC) concentration of 5 mM, which appeared to remain stable with higher concentrations of detergent (Figure 4A). The titration of the Fabs with increasing concentrations of peptide rendered comparable equilibrium dissociation constants (*K*_D_-s) and a 1:1 stoichiometry, indicating that the affinity for the helical epitope peptide solubilized in DPC micelles was unaffected by the potentiating mutations (Figure 4B and Table 1). In agreement with the previously reported values [8,11,16], these ITC assays yielded *K*_D_ values in the nM range. Interestingly, however, the binding energetics of the 2RF Fab showed a change in the thermodynamic signature, evidenced as a decrease in the Δ*H* component and a compensatory increase in the −*T*Δ*S* one. The higher contribution of the entropic component to binding in this mutant may reflect the Phe-induced rigidification of the HCDR3 loop before the Fab–MPER complex formation takes place [40]. 

We next measured the effects of the mutations on the Fab’s ability to bind to an MPER-TMD peptide that contained the full TMD moiety and that was reconstituted in the lipid membranes. Infrared spectroscopy confirmed the proper reconstitution of the peptide in membranes adopting a main α-helical conformation, as evidenced by a shift in the maximum of the spectrum from 1623 cm^−1^, which can be associated with an aggregation disposition, to 1652 cm^−1^, the characteristic band of the α-helical structures [41] (Figure 5A). The reconstituted MPER-TMD peptide presented the MPER epitope at the membrane surfaces in a form accessible for antibody binding, as judged from the promotion of the Fab–membrane association observed upon its inclusion in GUVs (Figure 5B and Appendix A). However, in this case a significant enhancement of the peptide-dependent binding to membranes was observed in the 2RF samples in comparison with the samples containing Fabs WT or 2R. The higher binding-affinity observed for the 2RF Fab mutant was confirmed by surface plasmon resonance (SPR) measurements, using supported lipid bilayers that contained reconstituted MPER-TMD peptides (Figure 5C and Appendix A). These binding kinetic measurements revealed that 2RF Fab binds its membrane-embedded peptide with a slower dissociation rate (*k*_off_), which likely contributes to its improved binding affinity and neutralization potency (Figure 2B and Figure 5D and Appendix A). 

Finally, we compared the binding affinity of the different 10E8 Fabs to native Env in intact virions using super-resolution stimulated emission depletion (STED) microscopy (Figure 5E) [5]. In this technique, the number of photons measured is directly proportional to the number of bound Fabs, allowing for the estimation and therefore comparison of the binding affinity of the different Fab mutants [42]. The results followed the same pattern observed in the binding (Figure 5D) and neutralization (Figure 2) experiments: the 2RF Fab signal showed the highest signal, followed by the 2R mutant, while the 10E8-WT Fab showed the lowest (Figure 5E). These data confirm that the increase in affinity towards reconstituted epitope peptides corresponded to an increase in its binding affinity for virion-associated Env trimers.

## 3. Discussion

The basis for the extraordinary neutralization breadth of 10E8 lies on the fact that this antibody recognizes the highly conserved amphipathic MPER helix [7,8,16,17]. This region is partially occluded by lipids, and consequently, 10E8 has evolved towards developing a lipid-accommodating surface to facilitate epitope recognition within the membrane environment [16,17,29,30,31]. Here, using antibody potency enhancement as a read-out, we have defined a protein–lipid interaction hotspot within the area defined by residues 30^LC^, 52^LC^ and 100c^HC^ that is directly involved in the biological function of 10E8. We hypothesize that the mutations within this Fab surface can improve neutralization potency and binding affinity through two non-mutually exclusive mechanisms. 

First, Arg-mediated electrostatic interactions might help to stabilize a neutralization-competent orientation of the Fab–MPER helix complex relative to the membrane (Figure 6). Thus, the combination of the Arg residues at positions 30 and 52 of the LC significantly augmented the neutralization potency of 10E8 in contrast to the lack of effect observed for each of these point mutations when combined with the S67R also in the LC of the antibody. Inspection of the orientation adopted by the MPER helix, when the Arg residues in each double mutant were placed in the plane of the lipid bilayer, showed that the antibody potentiation by S30R/N52R supported a “pole-like” model in which the MPER helix adopts an almost perpendicular orientation with respect to the membrane (Figure 6, top panel). In addition, this orientation reduces the angle of approach and increases the surface of the Fab in contact with the membrane probably reaching a more stable configuration (Figure 6, bottom panel).

Hence, it is likely that the presence of Arg-s 30 and 52 in close proximity to the paratope promotes the interactions with the membrane that correctly orientates the antibody and favors recognition of the native conformation adopted by the MPER helix within the viral membrane-anchored Env complex. The fact that the addition of the S100cF mutation further improved neutralization potency could be in part due to further stabilization of this neutralization-competent orientation adopted by the 2R mutant. 

Second, the inclusion of the Arg and Phe residues might contribute in the antigen-binding interaction energy. A favorable association of the Fab to the membrane is electrostatically driven by attraction between the Arg residues and anionic lipids, followed by intercalation of the nonpolar Phe into the lipid bilayer [39,43,44]. The latter event resulted in lower off-rate values for MPER peptides reconstituted in supported bilayers, and better interaction with its native epitope in HIV-1 virions, as measured by STED microscopy. Hence, the improved potency may also be partially derived from a kinetic advantage provided by the mutations to allow prolonged binding, which might be of high relevance in the case of antibodies with difficult-to-access epitopes, such as 10E8.

### Concluding Remarks

Lipid–membrane interaction was proposed as the first event that takes place in the epitope binding of anti-MPER bnAbs [45], including 10E8 [9], in order to facilitate binding to the transiently exposed Env intermediate. However, our data suggest that lipid binding appears to play a more specific role than just approaching the antibody to the membrane to facilitate further antibody docking. The fact that the Arg substitutions only have an effect on the neutralization potency of 10E8 when located in close proximity to the paratope, but not when distant from it (i.e., 67R_LC_, and 153R_LC_ or 193R_LC_ of the constant chain), suggests that the lipid interaction might participate in the stabilization of the Fab–epitope complex, once this is formed. The lack of avidity observed when comparing the neutralization potency of 10E8 WT and the 2RF mutant in the Fab and IgG format would further support this assumption. 

These observations open the possibility that the membrane-interacting surfaces could be functional elements of other antibodies outside the HIV field. In this context, the integral membrane proteins represent one of the largest fractions of immunotherapeutic targets under clinical evaluation. In many instances, such as in the case of the tumor-associated antigen CD20, these targets are multipass membrane proteins with small accessible external epitopes that are exposed close to, or directly lying on, the membrane surface. Thus, upon epitope binding, the antibodies targeting these receptors are in close proximity to the lipid bilayer and hence, studying their membrane-interacting features could offer an opportunity to study their binding mechanism and improve their functional properties.

Overall, our experimental data support that the 10E8–membrane contacts driven by electrostatic and hydrophobic-at-interface interactions have a defined role in the molecular recognition of the 10E8 epitope, as previously suggested by structural predictions [16,17]. The identification of the hotspot sites that establish functionally meaningful interactions with integral membrane antigens could guide the design of antibodies targeting membrane-proximal epitopes with optimized properties. 

## 4. Materials and Methods

### 4.1. Reagents

The following peptides were used in this study, (i) *KKK*-^671^NWFDITNWLWYIKLFIMIVGGLV^693^-*KK* (MPER_671–693_) and *KKK*-^671^NAADITNWLWYIKLFIMIVGGLV^693^-*KK* (_mut_MPER_671–693_) for the biophysical characterization of 10E8 Fab variants. Mutations W672A and F673A abolish antibody binding and thus, this peptide was used as a control to test the binding specificity of the antibodies; (ii) *KKKK*-^664^DKWASLWNWFDITNWLWYIKLFIMIVG^690^-*KKKKK* (MPER_664–690_) to compare the thermodynamic binding profile of 10E8 Fab variants. This peptide contains residues belonging to the gp41 transmembrane domain (TMD) (underlined) that have been described to increase the 10E8 binding affinity (REF scientific reports). In spite of the presence of TMD residues, this peptide can be solubilized in the presence of DPC allowing an assessment of the binding affinities in the absence of lipid membranes; (iii) *KKK*-^671^NWFDITNWLWYIKLFIMIVGGLVGLRIVFAVLSVVNRVR^701^ (MPER-TMD_671–709_) to measure the binding affinity to the epitope in the context of lipid membranes by SPR and confocal microscopy. The presence of the full TMD (underlined) allows for the reconstitution of the 10E8 epitope in membrane vesicles. These peptides were synthesized in the C-terminal carboxamide form by solid-phase methods using 9-fluorenylmethoxy carbonyl (Fmoc) chemistry, purified by reverse-phase high-pressure liquid chromatography, and characterized by matrix-assisted time-of-flight mass spectrometry (purity of >95%). Goat anti-Human IgG (Fab-specific)—Alkaline Phosphatase (AP)—was obtained from Invitrogen. The fluorescent probe Abberior Star Red (KK114) was obtained from Abberior (Göttingen, Germany). The 1-Palmitoyl-2-oleoyl-sn-glycero-3-phosphocholine (POPC) and 1-Palmitoyl-2-oleoyl-sn-glycero-3-phosphoserine (POPS) were purchased from Avanti Polar Lipids (Birmingham, AL, USA). The N-(7-Nitro-benz-2-oxa-1,3-diazol-4-yl)-phosphatidylethanolamine fluorescent probe was obtained from Thermo Fisher Scientific (Waltham, MA, USA). 

### 4.2. Antibody Production, Characterization and Labeling 

Heavy (HC) and light chain (LC) sequences of the Fabs were cloned into plasmid pColaDuet and expressed in the *Escherichia coli* T7-shuffle strain. Recombinant expression was induced at 18 °C overnight with 0.4 mM isopropyl-D-thiogalactopyranoside (IPTG) when the culture reached an optical density of 0.8. Protein purification was performed as previously described [31]. Briefly, the cells were harvested and centrifuged at 8000× *g*, after which they were resuspended in a buffer containing 50 mM HEPES (pH 7.5), 500 mM NaCl, 35 mM imidazole, DNase (Sigma-Aldrich, St. Louis, MO, USA) and an EDTA-free protease inhibitor mixture (Roche, Madrid, Spain). After cell lysis and cell debris removal, the Fabs were purified by a three-step process involving a nickel-nitrilotriacetic acid (Ni-NTA) affinity column (GE Healthcare) and a MonoS ion-exchange chromatography (IEC) column (GE Healthcare, Chicago, IL, USA). Purified Fab was concentrated and dialyzed against a buffer containing 10 mM sodium phosphate (pH 7.5), 150 mM NaCl, and 10% glycerol. For the preparation of the mutant Fabs, the KOD-Plus mutagenesis kit (Toyobo, Osaka, Japan) was employed following the instructions of the manufacturer. For the confocal microscopy experiments, STAR RED probe was conjugated at position C216_HC_ of the Fabs, as previously described [5].

LC and HC of 10E8v4 IgGs were cloned in the pHLsec expression vector and produced in HEK293-F cells. A total of 20 μg of the LC plasmid was co-transfected with 40 μg of the HC into 200 mL HEK293-F cells using PEIpro^®^ (Polyplus Transfections) at a 1:3 ratio of DNA: PEIpro. Cells were transfected at a cell density of 0.8 × 10^6^ cells/mL and incubated in an orbital shaker at 37 °C, 125 rpm and 8% CO_2_ for 7 days. The cells were harvested and supernatants retained and filtered with a 0.22 μm membrane (EMD Millipore). Supernatants were flowed through a protein A affinity column (GE Healthcare) using an AKTA Start chromatography system (GE Healthcare). The column was washed with 20 mM Tris pH 8.0, 150 mM NaCl and eluted with 100 mM glycine pH 2.2. Eluted fractions were immediately neutralized with 1 M Tris-HCl pH 9.0. The fractions containing protein were pooled, concentrated and flowed on a Superdex 200 Increase gel filtration column (GE Healthcare) to obtain purified samples, and stored in a buffer containing 10 mM sodium phosphate (pH 7.5), 150 mM NaCl and 10% glycerol.

### 4.3. Circular Dichroism 

Circular Dichroism (CD) measurements were obtained from a thermally controlled Jasco J-810 circular dichroism spectropolarimeter calibrated routinely with (1S)-(+)-10-camphorsulfonic acid, ammonium salt. Fabs were diluted to 5 μM in a PBS + 10% glycerol buffer, and spectra measured in a 1 mm path-length quartz cell initially equilibrated at 25 °C. Data were taken with a 1 nm band-width at 100 nm/min speed, and the results of ten scans were averaged.

### 4.4. Enzyme-Linked Immunosorbent Assay (ELISA)

The 96-well plates were coated overnight (ON) at room temperature (RT) with 1.37 μM of MPER derived peptide (KKK-_671_NWFDITNWLWYIKLFIMIVGGLV_693_-KK). A peptide with alanine mutations of the two underlined critical residues in the epitope was used as negative control. After a 2-h well blocking with 3% (w/v) bovine serum albumin (BSA), the serial dilutions of the Fabs (starting at 10 μg/mL) were incubated 1 h at RT. The bound Fabs were detected with an AP-conjugated goat anti-human immunoglobulin. The reaction was measured by absorbance at a wavelength of 405 nm in a Synergy HT microplate reader.

### 4.5. Pseudovirus Production

The functional screening of the Fab mutants was carried out in pseudovirus (PsV)-based cell-entry assays [46]. HIV-1 PsVs were produced by transfection of human kidney HEK293-T cells with the full-length Env clone JRCSF (kindly provided by Jamie K. Scott and Naveed Gulzar, Simon Fraser University, BC, Canada) and the PV0.4 molecular clone (obtained from the AIDS Research and Reference Reagent Program (ARRRP)). The cells were co-transfected with vectors pWXLP-GFP and pCMV8.91, encoding a green fluorescent protein and an Env-deficient HIV-1 genome, respectively (provided by Patricia Villace, CSIC, Madrid, Spain). For STED experiments, the vectors pCMV8.91 Env, pmCherry-Vpr and the PV0.4 Env clone were used for transfection. The pmCherry-Vpr and PVO clone 4 vectors were obtained through the NIH HIV Reagent Program, Division of AIDS, NIAID, NIH: Human Immunodeficiency Virus-1 VAI Vpr Expression Vector, ARP-12479, contributed by Dr. Thomas J. Hope; Vector pcDNA3.1 D/V5-His TOPO© Expressing Human Immunodeficiency Virus Type 1 (HIV-1) Env [PVO, clone 4 (SVPB11)], ARP-11022, contributed by Dr. David Montefiori, Dr. Feng Gao and Dr. Ming Li. After 24 h, the medium was replaced with Optimem-Glutamax II (Invitrogen Ltd., Paisley, UK) without serum. Two days after transfection, the PsV particles were harvested, passed through 0.45 μm pore sterile filters (MillexHV, Millipore NV, Brussels, Belgium) and finally concentrated by ultracentrifugation in a sucrose gradient. For the STED experiments, virions were concentrated using the Lenti-X Concentrator reagent (Takara Bio, Shiga, Japan).

### 4.6. Cell-Entry Inhibition Assay

HIV-1 entry was determined using CD4^+^CXCR4^+^CCR5^+^ TZM-bl target cells (ARRRP, contributed by J. Kappes). HIV-1 PsVs were first diluted to 10–20% tissue culture infectious doses in DMEM supplemented with inactive serum, and added to decreasing concentrations of the 10E8 WT and mutant Fabs. Infection levels after 72 h were inferred from the number of GFP-positive cells as determined by flow cytometry using a CytoFLEX flow cytometer (Beckman Coulter, Brea, CA, USA).

### 4.7. Transmission Infrared Spectroscopy

The infrared spectra were recorded in a Thermo Nicolet Nexus 5700 (Thermo Fisher Scientific, Waltham, MA, USA) spectrometer equipped with a liquid nitrogen-refrigerated mercury-cadmium-telluride detector using 25μm-pathlength calcium fluoride cells (BioCell, BioTools Inc., Wauconda, IL, USA). MPER-TMD containing samples were lyophilized and subsequently prepared at 3 mg (peptide)/mL in D2O buffer (PBS). A 25 μL sample aliquot was deposited on the bottom window of the cell that was sealed with the top window. The reference samples without peptides were prepared similarly. Typically, 102 scans were collected for each background and sample, and the spectra were obtained with a nominal resolution of 2 cm^−1^.

The band decomposition of the original amide I were completed as previously described [41]. In brief, the number and position of bands were obtained after deconvolution (bandwidth = 18 and k = 2) and the Fourier transformation (power = 3 and breakpoint = 0.3) of the spectra. The baseline was removed before starting the fitting procedure and the initial heights set at 90% of those in the original spectrum for the bands in the wings and for the most intense component, and at 70% of the original intensity for the rest of the bands. An iterative process followed, in two stages. (i) The band position of the component bands was fixed, allowing the widths and heights to approach final values; (ii) the band positions were left to change. For band shape, a combination of Gaussian and Lorentzian functions was used. The restrictions in the iterative procedure were needed because the initial width and height parameters can be far away from the final result due to the overlapping of the bands, so that spurious results can be produced. In this way, the information from band position, percentage of amide I band area and bandwidth were obtained for every component. Using this procedure, the result was repetitive. Mathematical accuracy was assured by constructing an artificial curve with the parameters obtained and subjecting it to the same procedure again. The number of bands was fixed on the basis of the narrowing procedures. The molar absorption coefficient for the different bands was assumed to be similar and within a +/−3% error.

### 4.8. Single-Vesicle-Binding Assay

Giant unilamellar vesicles (GUVs) were produced following the electro-formation method as described in previous works (Apellaniz et al., 2010). A total of 2 mM of lipid (POPC or POPC:POPS (1:1)) was dissolved in 200 μL CHCl_3_:CH_3_OH with the fluorescent probe NBD-PE (0.5%). When required, MPER-TMD_671–709_ peptide (dissolved in 10% (v/v) 1,1,1,3,3,3-hexafluoro-2-propanol (HFIP)) was included in the organic phase at 1:250 peptide-to-lipid ratio. The GUVs were transferred to a BSA-blocked microscope chamber and incubated for 15 min with 250 nM of 10E8-STAR RED fluorescent Fabs. The images were acquired on a Leica TCS SP5 II microscope (Leica Microsystems GmbH, Wetzlar, Germany). NBD-stained GUVs were excited at 476 nm, and emission was imaged at 530 ± 20 nm by using a 63× water immersion objective (numerical aperture (NA) = 1.2). The STAR RED-labeled Fab fragments were excited at 633 nm by using a HeNe laser, and emission was imaged at 775 ± 125 nm. Relative extents of Fab–GUV binding were obtained by measuring the fluorescence intensity of STAR RED along the equatorial plane of the GUV images, and normalized with WT values as the reference, in a number of vesicles *n* ≥ 8 and *n* ≥ 17 in lipid alone and lipid–peptide samples, respectively.

### 4.9. Isothermal Titration Calorimetry (ITC)

The stoichiometry (n) and binding constants (K_D_) of 10E8 binding to the MPER_664–690_ peptide *KKKK*-^664^DKWASLWNWFDITNWLWYIKLFIMIVG^690^-*KKKKK* was measured by a VP-ITC microcalorimeter (MicroCal, Northampton, MA, USA) at 25 °C. All of the samples used in the ITC experiments were dialyzed against a buffer containing 10 mM sodium phosphate (pH 7.5), 150 mM NaCl, supplemented with 10% glycerol and 5 mM DPC. Fabs at 3 μM were titrated with 12 injections of 40 μM peptide. Curve fitting to a one-site binding model was performed with ORIGIN 7.0 software (MicroCal, Northampton, MA, USA).

### 4.10. Surface Plasmon Resonance (SPR) Measurements

Large unilamellar vesicles (LUVs) were produced as follows. 1 mM of lipid (POPC) was diluted in chloroform: Methanol in a 1:2 vol/vol ratio and the MPER-TMD_671–709_ peptide (dissolved in 10% HFIP) was added to the organic phase at 1:1000 peptide-to-lipid ratio. The mixture was dried under a nitrogen stream and the organic traces were removed using a high vacuum for 1 h. The lipid film was pre-hydrated for 30 min using a bubbler under humidified nitrogen conditions. Then, the lipid film was dissolved in 1ml HBS using a sonication bath at 65 °C for 1 h and LUVs were formed following a freeze–thaw cycling process.

SPR assays were carried out in an MP-SPR instrument SPR Navi 220-NAALI (BioNavis, Tampere, Finland) using low-molecular-weight (6 kDa) dextran-coated sensors attached to lipophilic groups that promoted the formation of supported lipid bilayers (SLBs) [47]. After lipid binding, a short pulse of 100 mM NaOH in HBS (buffer prepared from ready-made tablets (Medicago, Uppsala, Sweden)) was used to detach the loosely bound lipids and 0.1% BSA in HBS (150 mM NaCl, 10 mM HEPES pH 7.4) to block the chip surface. Kinetic measurements of binding of Fabs were performed at 25 °C in HBS with a fast flow rate (100 µL/min) using SLBs that contained a low peptide membrane density (peptide-to-lipid ratio, 1:1000), so that the mass-transfer effects became negligible. Moreover, in comparison with the supported vesicles, the SLB format maximized the epitope exposure, and minimized the potential steric effects (constrained accessibility) on the SPR sensor slide. The peptide-bound Fabs were later stripped with 100 mM NaOH in HBS. In these experiments, the reference channel was a second flow chamber of the same chip covered with an SLB devoid of peptide. The sample was passed through both channels, and the signal was automatically subtracted by the program SPR-Navi Data Viewer: 6.6.3.1. (Bionavis, Tampere, Finland). *k*_on_ and *k*_off_ values were calculated by fitting the kinetic sensograms to a one-to-one model using the software TraceDrawer 1.9.2 (Ridgeview Instruments AB, Uppsala, Sweden), and the equilibrium dissociation constant (*K*_D_) was automatically calculated by the program.

### 4.11. STED Imaging

Vpr-mCherry-labelled virions were immobilized on 0.01% poly-L-lysine-coated 18 mm coverslips for 10 min. The coverslips were blocked with 2% fatty-acid free bovine serum albumin (BSA) for 30 min and subsequently incubated with 0.5 μM Fab in 1% fatty-acid free BSA. The samples were fixed with 2% paraformaldehyde, mounted in 7 μL 2.5% Mowiol/DABCO and imaged 24 h after mounting. 

STED images were acquired using an Abberior STEDYCON microscope (Abberior Instruments, Göttingen, Germany) attached to an Olympus IX83 inverted microscope body (Olympus, Tokyo, Japan) equipped with a 100×/1.40 NA UPlanSApo Olympus objective lens. mCherry was excited with a 561 nm laser and STAR RED with a 640 nm laser. The laser power for both of the laser lines at the sample plane was 10 μW. The depletion to achieve the STED effect was performed with a doughnut-shaped 775 nm laser beam, with a mean 100 mW measured power at the sample plane. The emission was collected using photon-counting avalanche photodiodes. Pixel size was 30 nm, dwell time 10 μs and the signal from five subsequent line scans was integrated for the STED channel. Image analysis was performed using a python script developed for this purpose [5,48] available at the Zenodo repository (DOI:10.5281/zenodo.1465920). The virus regions of interest (ROIs) were identified on the confocal mCherry channel using a maximum intensity finding algorithm on a Gaussian 2-pixel smoothed image. Quantification of the Fab STED signal, i.e., detected photons, was performed on 10 pixel-diameter circular regions centered around the mCherry-Vpr intensity maxima, i.e., corresponding to single virions. To compare the data from independent experiments, the signal was normalized to the median intensity signal of the 10E8-WT Fab. A total of 233 (WT), 156 (2R) and 278 (2RF) virions were measured in three independent experiments.

## Figures and Tables

**Figure 1 ijms-23-10767-f001:**
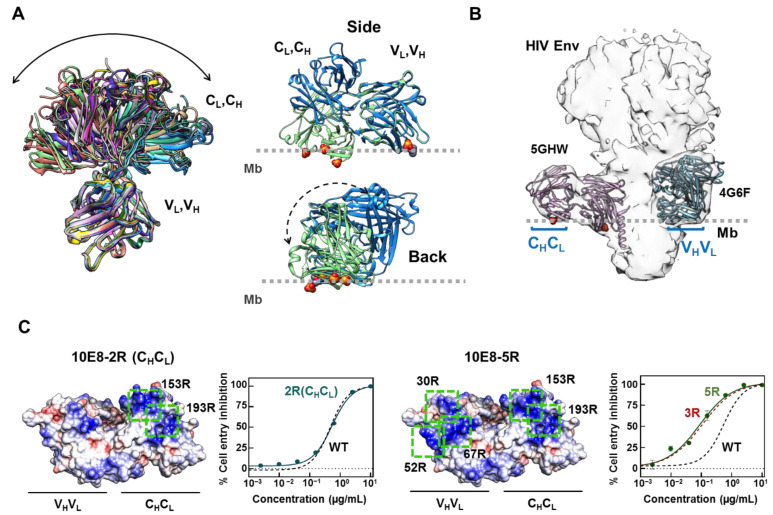
Delineation of membrane-facing/solvent-exposed surfaces in the Fab 10E8. (**A**) Left: superposition of a set of 10E8 Fab atomic structures with different elbow angles between the variable and constant domains. Superposed structures with PDB IDs 4G6F, 4U6G, 5GHW, 5IQ7, 5JNY, 5SY8, 5T29, 5T5B, 5T6L, 5T80, 5T85 and 5TFW are depicted. The V_H_V_L_ domain was used for structure alignment. Right: Position relative to the membrane of Fab 10E8 structures with PDB ID 5GHW (green) and 5T29 (blue). The structures have been aligned through the V_H_V_L_ domain and oriented based on the bound phosphates (red spheres) present in the former structure. The back side of the antibody emphasizes the possibility of the C_H_C_L_ domain rotation to make contact with the membrane (Mb). (**B**) Fitting into Cryo-EM map of a Fab–Env trimer complex determined in detergent micelles (EMD-3312). Two Fabs, exhibiting full occupancy and different orientations are clearly discernable (based on Fab structures 5GHW and 4G6F, as indicated in the panel). Membrane-facing areas hosting the Arg substitutions in both the variable (V_H_V_L_) and constant (C_H_C_L_) regions are highlighted. (**C**) Pseudovirus neutralization and surface density charge representation of Fab 10E8 (PDB ID: 5GHW) with residues 153S and 193S in the constant region mutated to Arg alone, labeled as 10E8-2R (C_H_C_L_) (left panels), or in combination with mutations S30R, N52R and S67R of Fab-3R labeled as 10E8-5R (right panels). The positions of the Arg mutations are highlighted with green boxes. Neutralization curves for 10E8-WT and 10E8-3R are shown for comparison. Structures displayed in the panels rendered with Chimera [38].

**Figure 2 ijms-23-10767-f002:**
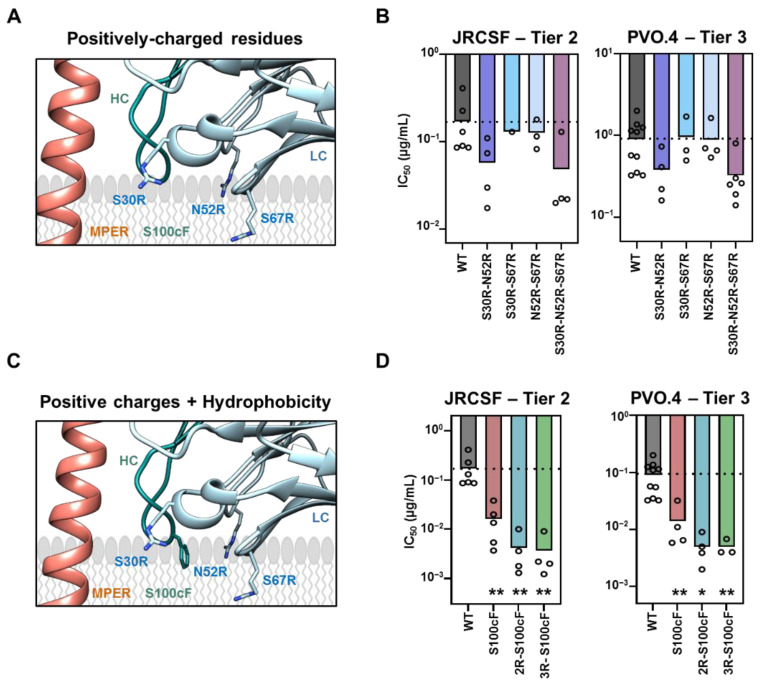
Contribution of charge and hydrophobicity-at-interface in the neutralization of 10E8**.** (**A**) Cartoon representation depicting the position of the three Arg-s at the membrane-facing surface of the antibody (blue) with respect to the membrane (gray) and its epitope peptide (orange). (**B**) IC_50_ comparison of 10E8 Fab variants (different shades of blue) and 10E8-WT (dark gray) with double and triple combinations of Arg-s against JRCSF and PVO.4 PsV (mean values ± SD for a minimum of three experimental replicates). (**C**) Position of mutation S100cF (green) relative to the three Arg-s in 10E8-3R. Color code as in A. (**D**) Effect of adding the mutation S100cF in the neutralization potency of 10E8 (red), 10E8-2R (blue) and 10E8-3R (green). * and ** indicate significance compared to 10E8-WT (*p* < 0.1 and *p* < 0.01, respectively) by ANOVA.

**Figure 3 ijms-23-10767-f003:**
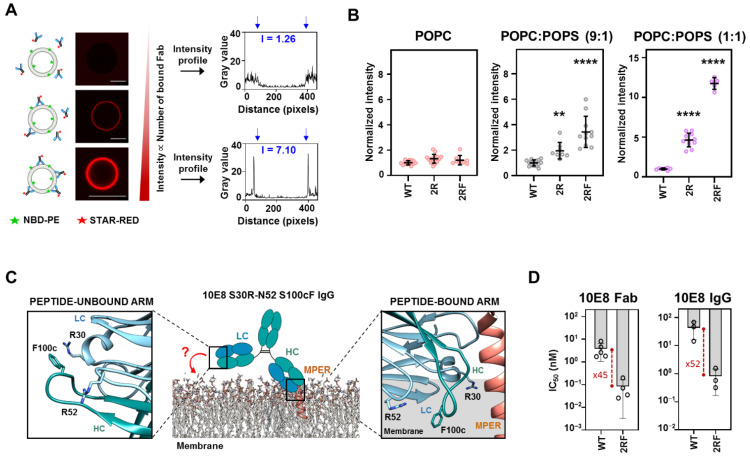
Contribution of the Arg-s and S100cF mutations in the direct binding of 10E8 to lipid membranes. (**A**) Scheme to illustrate Fab-binding quantification method to lipid vesicles. Fabs are labeled with the fluorescent dye STAR RED to lipid membranes. The lipid NBD-PE was used to localize the vesicles and the fluorescence intensity of STAR RED-Fabs at the lipid bilayer section (indicated with blue arrows) was recorded. Scale bars, 5 μm. (**B**) Comparison of the binding intensities of 10E8 variants to vesicles (number of vesicles *n* ≥ 8) with increasing amount of the anionic phospholipid POPS (0% pink, 10% gray and 50% purple). ** and **** indicate significance compared to 10E8-WT (*p* < 0.01 and *p* < 0.0001, respectively) by ANOVA. (**C**) Schematic cartoon to illustrate the possible avidity-based potency enhancement of IgG molecules through direct membrane binding. (**D**) IC_50_ increase comparison against JRCSF PsV between mutated and 10E8-WT when displayed as monovalent Fabs or as bivalent IgG molecules.

**Figure 4 ijms-23-10767-f004:**
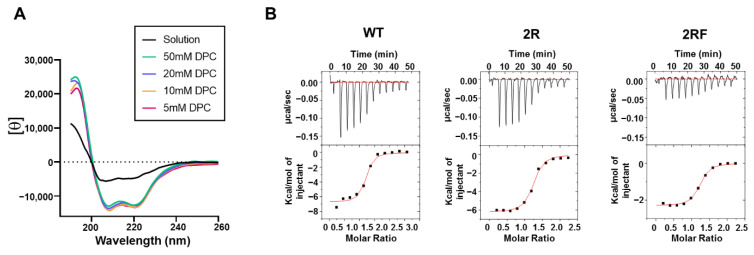
Binding comparison of helical MPER epitope in solution. (**A**) CD spectra of MPER_664–690_ peptide in solution and increasing concentration of DPC micelles as indicated in the panel. (**B**) Binding isotherms of the peptide solubilized in 5 mM DPC to 10E8-WT, 10E8-2R and 10E8-2RF Fabs. Upper panels: heat released upon peptide–Fab interaction. Lower panels: Integrated heats (symbols) and non-linear least-squares fit (line) of the data to a one-site binding model. The thermodynamic binding parameters are shown in Table 1.

**Figure 5 ijms-23-10767-f005:**
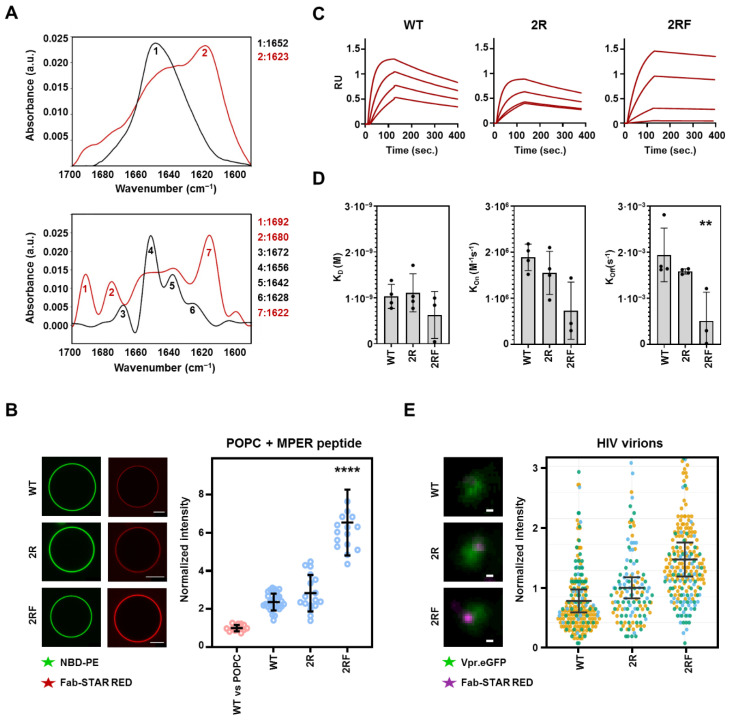
Binding comparison of MPER epitope reconstituted in lipid membranes**.** (**A**) Top: Infrared spectra of MPER-TMD_671–709_ peptide in PBS (red) and reconstituted in POPC vesicles (black) in the amide I region. Bottom: after deconvolution the spectra display the different components of the absorption bands. Wavelengths corresponding to the main peaks (labeled 1–7) are displayed for clarity. (**B**) Representative (left) and quantitative (right) binding of 10E8 variants to POPC vesicles containing reconstituted MPER-TMD_671–709_ helices (number of vesicles *n* ≥ 17). The 10E8-WT binding to vesicles without peptides was added for reference (red dots). Scale bars, 5 μm. **** indicate significance compared to 10E8-WT Fab (*p* < 0.0001) by Kruskal–Wallis test. (**C**,**D**) Sensograms (**C**) and kinetic parameters (**D**) of 10E8 Fab variants binding to supported lipid bilayers that contained reconstituted MPER-TMD peptides obtained by surface plasmon resonance (SPR). ** indicates significance compared to 10E8-WT Fab (*p* < 0.01) by one-way ANOVA. (**E**) Left, STED super-resolution images of STAR RED-labelled Fabs (magenta) bound to native Env in individual virions labelled with Vpr.mCherry (green). Scale bars, 100 nm. Right, quantification of the Fab signal from individual HIV-1 virions from STED super-resolution images. Measured photon counts were normalized to the median signal of the 10E8-WT Fab. Each dot represents the signal from an individual virion (n_WT_ = 233, n_2R_ = 156, n_2RF_= 278) and colors represent each of three independent experiments performed.

**Figure 6 ijms-23-10767-f006:**
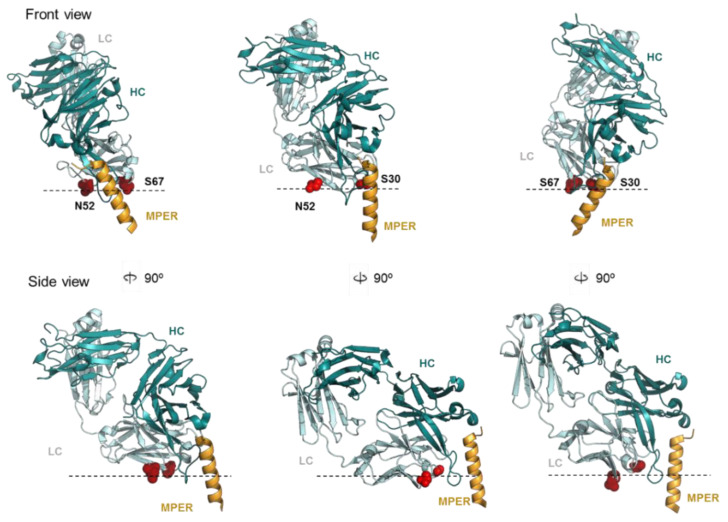
Binding models of 10E8 Fab to the helical MPER epitope inserted in membranes**.** The panels display front (top) and side views (bottom) of the Fabs oriented relative to the membrane (dotted lines) according to the Arg positions in the 2R mutants. Two features define the combination of S30R/N52R that enhances potency of the antibody (middle models): (i) the MPER helix with its main axis almost perpendicular to the membrane plane; and (ii) an adjacent LC β-sheet establishing full contact with the membrane surface.

**Table 1 ijms-23-10767-t001:** Thermodynamic parameters of binding of Fab 10E8 and its mutants to the peptide MPER_664–690_.

Fab	*K*_D_ (nM)	*n*	Δ*H* (kcal mol^−1^)	−*T*Δ*S* (kcal mol^−1^)
WT	32.39	1.12	−6.73	−3.49
2R	46.29	1.17	−6.24	−3.75
2RF	41.3	1.16	−2.3	−7.77

## Data Availability

The data that support the findings of this study will be available from the corresponding author upon reasonable request.

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
