# Peer review of "Functional Delineation of a Protein–Membrane Interaction Hotspot Site on the HIV-1 Neutralizing Antibody 10E8"

_ijms, 2022, doi:10.3390/ijms231810767_

Round 1
Reviewer 1 Report
The manuscript entitled: "Functional Delineation Of A Protein-Membrane Interaction Hotspot Site On The Hiv-1 Neutralizing Antibody 10e8" submitted by Sara Insausti, Miguel Garcia-Porras, Johana Torralba, Izaskun Morillo, Ander Ramos-Caballero, Igor de la Arada, Beatriz Apellaniz, Jose M.M. Caaveiro, Pablo Carravilla, Christian Eggeling, Edurne Rujas, and Jose L. Nieva, is interesting, nicely written, and it can be published in Int. J. Mol. Sci., but after minor revisions, which are listed below.
1. The novelty of the studies should be emphasized more.
2. The HFIP abbreviation should be explained
3. A comparison of the Kd value with Kd values obtained for other MPER peptides/systems in the literature should be presented/commented on.
4. In the Materials and Methods section, a few peptides were synthesized. Their role is unclear. Please clarify it. Why have the Authors presented the thermodynamic parameters of binding of Fab 10E8 and its mutants to only the peptide MPER664-690? Please clarify it. What was the application of one of the synthesized peptides, the mutMPER671-693.
Reviewer 2 Report
This manuscript presents a detailed multidisciplinar analysis trying to correlate structure, membrane-binding capabilities, and potency as antibodies able to recognize a defined HIV epitope either isolated or within the context of HIV-like virions, in order to understand some basic principles of the mechanisms of an interesting family of broadly neutralizing anti-HIV antibodies. The conclusion is that the structure of these antibodies may present a particular region optimized to interact with the membranes where the targeted epitopes are inserted. This opens interesting lines of research to optimize further these and other possible antibodies that could have therapeutic interest.
The study joins an impressive collection of advanced methodologies, which have been applied in a complementary and smart way. The experiments seem to have been carried out carefully and the results are explained in a very clear manner.
I only have a few minor questions that the authors may like to address to improve the paper.
Although the reported membrane-interacting features have been described and analyzed in the specific context of a well-defined family of antibodies, could he authors discuss whether features in this line could also define the activity, in general, of antibodies targeting membrane-associated epitopes?
The quality introduced by the inclusion of a Phe residue in the S100cF mutation is taken as “hydrophobicity”, as providing hydrophobic interactions, as Arg’s introduce electrostatic interactions. I do not completely agree with this. If hydrophobicity would be the critical property, a mutation to introduce other large strictly hydrophobic residues (Leu, Val…) may also work. Would not be “aromaticity” the real favorable property? Aromatic residues have been described to have favorable partition into phospholipid membrane headgroup interfaces. Would a Tyr residue make the same work?
I am not sure whether one can talk about true membrane “insertion” of the tested Fabs (i.e. in line 206). The authors have strictly tested whether Fabs interact or associate with membranes. To what extent this is a “insertion”, understood as a true intercalation at some depth between the phospholipid molecules, has not been determined, at least here.
What has been tested in the illustrative images shown as reference in Figure 3A?
Please, provide in the figures and in materials and methods the number of total vesicles, or virion particles, that have been measured to obtain the given numbers.
Please, indicate in the legend of Figure 5, that data presented in the panel E have been obtained by SPR.
Please, correct the degree symbol along the text (i.e. in line 111, or line 407), distinct from the “º” superindex.
The end of the “STED imaging” methodological subsection copy a fragment of the legend of Figure 5.
Author Response
Please, see the attachment
